# Correlations of Sesamoid Bone Subluxation with the Radiologic Measures of Hallux Valgus and Its Clinical Implications

**DOI:** 10.3390/medicina59050876

**Published:** 2023-05-02

**Authors:** Sung Hwan Kim, Young Hwan Kim, Joo Young Cha, Young Koo Lee

**Affiliations:** Department of Orthopaedic Surgery, Soonchunhyang University Bucheon Hospital, 170, Jomaru-ro, Wonmi-gu, Bucheon-si 14584, Gyeonggi-do, Republic of Korea; shk9528@naver.com (S.H.K.); remedios@schmc.ac.kr (Y.H.K.); cacarito@hanmail.net (J.Y.C.)

**Keywords:** hallux valgus, hallux valgus angle, intermetatarsal angle, joint congruency, sesamoid bone, sesamoid bone subluxation

## Abstract

*Background and Objectives*: Hallux valgus is one of the most common chronic foot complaints, with prevalences of over 23% in adults and up to 35.7% in older adults. However, the prevalence is only 3.5% in adolescents. The pathological causes and pathophysiology of hallux valgus are well-known in various studies and reports. A change in the position of the sesamoid bone under the metatarsal bone of the first toe is known to be the cause of the initial pathophysiology. *Purpose*: The relationships between the changes in the location of the sesamoid bone and each radiologically measured angle and joint congruency in the hallux valgus remain as yet unknown. Therefore, this study investigated the relationships of sesamoid bone subluxation with the hallux valgus angle, intermetatarsal angle, and metatarsophalangeal joint congruency in hallux valgus patients. The goal is to know the hallux valgus angle, the intermetatarsal angle, and metatarsophalangeal joint congruency’s correlation with hallux valgus severity and prognosis by revealing the relationship between each measured value and sesamoid bone subluxation. *Materials and Methods*: We reviewed 205 hallux valgus patients who underwent radiographic evaluation and subsequent hallux valgus correction surgery in our orthopedic clinic between March 2015 and February 2020. Sesamoid subluxation was assessed using a new five-grade scale on foot radiographs, and other radiologic measurements were assessed, such as hallux valgus angle, the intermetatarsal angle, distal metatarsal articular angle, joint congruency, etc. *Conclusions*: Measurements of the hallux valgus angle, interphalangeal angle, and joint congruency exhibited high interobserver and intraobserver reliabilities in this study. They also showed correlations with sesamoid subluxation grade.

## 1. Introduction

Hallux valgus is one of the most common chronic foot complaints, with reported prevalences of over 23% in adults, 35.7% in older adults, and only 3.5% in adolescents [1,2]. Hallux valgus recurrence after corrective surgery is a well-known phenomenon; the long-term recurrence rate can reach up to 50%. The causes of recurrence are thought to be multifactorial, including surgical factors such as choice of the appropriate procedure and technical competency and patient-related factors such as anatomic predisposition, medical comorbidities, and compliance with post-correction instructions [3]. The etiology of hallux valgus is multifactorial; both intrinsic and extrinsic factors may be involved, and the condition tends to be inherited [2]. Hypermobility of the first ray began to be considered as an etiologic factor and related as a primary cause of hallux valgus [4]. Hallux valgus can make it difficult for patients to wear fashionable shoes; it may also impair their quality of life by restricting daily and recreational activities [3]. Difficulties associated with hallux valgus include foot pain, impaired balance, awkward gait pattern, and fall down, especially in older adults [5]. Flat foot and navicular bone drop are typical symptoms of hallux valgus. Patients may experience gradual changes in the alignment and shape of the forefoot and midfoot, and the changes frequently occur in the medial direction [6]. Medial deviation of the first metatarsal bone is common, along with lateral deviation of the first phalange, deformity of the phalangeal bone and interphalangeal joint, and pronation of the big toe in conjunction with sesamoid subluxation [7]. Radiologic studies showed that the metatarsophalangeal joint changes into a curved shape in hallux valgus patients. The first metatarsal bone’s length also increases. Abnormal alignment and structure of the first metatarsophalangeal joint would contribute to the collapse of the medial longitudinal arch of the foot in hallux valgus patients [8]. Intense plantar pressure below the hallux and first metatarsal area are well-known biomechanical characteristics of hallux valgus [9]. Zhang et al. found that the metatarsal areas exhibited more stress in the hallux valgus patients, especially the first metatarsal area, in a finite element study that dealt with the subject of metatarsal stress and metatarsophalangeal loadings between hallux valgus patients and a healthy control group. Moreover, foot kinematics analyzed in a multi-segmental study resulted in differences between hallux valgus patients and the healthy control group [10]. Treatment options for hallux valgus deformity comprise surgical treatment and conservative treatment [11]. Although more than 100 open surgical methods are available, there is no clear consensus regarding which is the most effective option [12]. To name just a few examples, in surgical treatments, Akin osteotomy, the first metatarsal osteotomy, the McBride procedure and the Lapidus procedure are widely used surgical options for moderate or severe hallux valgus patients with discomfort [13]. Percutaneous approaches and minimally invasive surgery are increasingly used because these approaches can achieve results that are at least equivalent to the results of conventional open surgery and have lower complications [14,15]. Conservative treatment is also recommended for specific patient groups. In previous studies, it was proven that foot orthosis could lower peak pressure loadings of the hallux valgus patients’ feet [16,17]. Lee et al. *. and Karabicak et al. found that kinesiology tape for the foot could relieve foot pain and decrease the hallux valgus angle [18,19]. Less frequent jogging and the avoidance of shoes can be beneficial for the intrinsic muscles of the foot, thus promoting healthy foot arch development and gait [20].

Evaluation of hallux valgus deformity via conventional radiography provides surgeons with the necessary information to choose the correct treatment option and appropriate surgical procedure if the patient needs surgery. Considering those facts, systematic radiographic evaluation of hallux valgus deformity is important for the achievement of good surgical outcomes [21]. However, most radiographic measurements focus on the angular deformities in the transverse plane, which are measured on dorsoplantar foot radiographs [5]. Rotational deformities in the coronal plane have attracted less attention, although they are likely to affect those in the coronal plane [22]. Radiological measurements, including the hallux valgus angle, the intermetatarsal angle, the distal metatarsal articular angle, and the first metatarsophalangeal joint congruency, are used to determine deformity etiology, grade, and extent. Almost all of these radiographic measurements are used to identify etiology, grade the extent of deformity, and decide the treatment plan and its surgical method if needed [23]. The value of hallux valgus angle, intermetatarsal angle and extent of sesamoid displacement are used to classify hallux valgus deformity and help to derive treatment algorithms [9,24]. The value of the hallux valgus angle on plain foot X-rays is an important predictor of the outcome of hallux valgus correction surgery [25]. Hallux valgus deformity is usually indicated by sesamoid subluxation beginning in the first metatarsal head [26]. In a study by Ryuhei et al., it was demonstrated that the sesamoid bone’s lateral displacement is strongly associated with the severity of hallux valgus [27]. Okuda et al. demonstrated whether reduction of the sesamoid bone to the first metatarsal head is completed or not would be an important component in the correction surgery of hallux valgus because the incomplete reduction of the sesamoid bone may cause the deformity to recur postoperatively [28]. Considering that knowledge, precise and detailed assessment of the sesamoid bone is important to ensure that the most appropriate treatment option for hallux valgus deformity is selected. The sesamoid bone of the first metatarsophalangeal joint has several functions. For example, it absorbs most weight on the first ray; this protects the flexor hallucis longus tendon, which courses over the first plantar surface of the metatarsal head and enhances the mechanical function of the intrinsic muscles of the first ray [9]. The intersesamoid ligament, which is under the first metatarsal head, contributes to the intrinsic stability of the sesamoid complex.

Foot weight-bearing computer tomography (CT) has become an accurate and highly valuable radiological method to assess several foot and ankle diseases [29,30,31]. Collan et al. were the first to use weight-bearing CT for the assessment of hallux valgus patients [32]. Hallux valgus measurements obtained via weight-bearing CT are highly reliable. In particular, the distal metatarsal articular angle can be measured with high accuracy. However, Zhong et al. do not recommend foot weight-bearing CT for all hallux valgus patients because not every hallux valgus patients need to take a foot weight-bearing CT scan [8]. In fact, it is not easy for clinicians to use such a method because not many hospitals have foot-weight-bearing CT equipment. Considering issues such as cost, ease of inspection, and radiation exposure, a plain radiograph is still an attractive option that cannot be ignored. Even considering the diagnostic aspect, except for the distal metatarsal articular angle, some parameters (hallux valgus angle, intermetatarsal angle, proximal phalangeal articular angle, and sesamoid subluxation) measured by plain radiographic are comparable to weight-bearing CT [6].

So far, there are many studies that report the reliability of various radiographic measurements evaluating a hallux valgus deformity with a plain radiograph [33,34,35]. However, to our knowledge, the correlation between sesamoid bone subluxation and hallux valgus angle, the intermetatarsal angle, and metatarsophalangeal joint congruency in hallux valgus patients with foot plain radiograph have not been described in English-language medical literature. We hypothesized that the degree of subluxation of sesamoid bone might have a relationship with radiologic measurement and severity of the hallux valgus. The purpose of our study is to establish the reliability of eight radiologic measurements and to determine the relationship of sesamoid subluxation with other radiologic measurements with a plain foot radiograph.

## 2. Materials and Methods

We retrospectively reviewed 205 hallux valgus patients who underwent radiographic evaluation and subsequent hallux valgus correction surgery in our orthopedic clinic between March 2015 and February 2020. The radiographic evaluation included weight-bearing dorsoplantar and lateral foot radiographs. We excluded 25 patients, including 10 who only had non-weight-bearing scans, four with brachymetatarsia, three with cavus foot deformity, three with Charcot arthritis, two with claw toe deformity, one with bunionette deformity, one with gouty arthritis, and one with crushing injury (Figure 1). These patients were excluded because non-weight-bearing radiographs cannot clearly show hallux valgus deformity, and combined diseases like inflammatory arthritis, trauma, and other toe deformities can influence the hallux valgus condition. In total, images of both feet of 180 patients were included in the final analysis. The mean patient age was 52.78 years (range: 15–78 years; 18 men and 162 women) (Table 1). The study was conducted in accordance with the Declaration of Helsinki, and approved by the Institutional Review Board and Human Research Ethics Committee of Soonchunhyang University Bucheon Hospital (IRB No. 2023-02-015-001, 20 March 2023).

Foot radiographs were acquired using the Innovision-SH instrument (DK Medical Systems, Seoul, Republic of Korea; 50 kVp, 5 mAs) at a distance of 100 cm and with each patient standing upright. We retrieved the radiographic images using a picture archiving and communication system (PACS; DEJA-VIEW; Dongeun Information Technology, Bucheon, Republic of Korea). Radiographic measurements were performed using PACS 1.42 software. 

Eight radiological measurements were made, including seven on dorsoplantar foot radiographs (hallux valgus angle, intermetatarsal angle, distal metatarsal articular angle, proximal phalangeal articular angle, hallux interphalangeal angle, sesamoid subluxation, and congruency) and one on lateral view radiographs (tarso-first metatarsal angle [i.e., Meary’s angle]). On foot weight-bearing plain radiographs, the hallux valgus angle is the angle between the longitudinal axis of the first metatarsal bone and the longitudinal axis of the proximal phalanx bone; the intermetatarsal angle is the angle between the longitudinal axis of the first and second metatarsal bones; the distal metatarsal articular angle is the angle between a line perpendicular to the longitudinal axis of the first metatarsal bone and a line delineating the orientation of the articular surface of the metatarsal bone’s head; the proximal phalangeal articular angle is the angle between a line delineating the orientation of the base of the proximal phalangeal articular surface and a line delineating the orientation of the proximal phalangeal distal articular surface; and the hallux interphalangeal angle is the angle between the longitudinal axis of the proximal phalanx bone and the longitudinal axis of the distal phalanx of the hallux.

Sesamoid subluxation was assessed using a new five-grade scale, which can also describe the relationship between the tibial sesamoid and the longitudinal axis of the first metatarsal bone. A sesamoid, which had no lateral displacement relative to the bisection line, was deemed as grade 0. Grade 1 occurred when there was an overlap of less than 25% of the sesamoid to the bisection. Grade 2 was when the overlap of the sesamoid became greater than 25% and less than 50% of the bisection. Grade 3 was when the overlap of the sesamoid became greater than 50% and less than 75% of the bisection. Grade 4 was when the overlap of the sesamoid became greater than 75% (Table 2). The congruency of the metatarsophalangeal joint was determined based on the relationship between the articular surface of the base of the proximal phalangeal bone and the first metatarsal bone’s head (Figure 2). Tarso-first metatarsal angle (Meary angle) assessed the longitudinal arch of the medial column of the foot and was measured between the long axis of the talus drawn from the midpoint of the talar body through the mid-diameter of the talonavicular joint and the long axis of the first metatarsal bone (Figure 3).

Three orthopedic surgeons (a first-year resident, third-year resident, and foot and ankle fellow) independently performed the radiographic measurements. They were each blinded to the patients’ clinical information and measurements by the other surgeons. To determine intraobserver reliability, the three surgeons repeated measurements of 240 radiographs after an interval of two weeks. We assessed interobserver and intraobserver reliabilities using intraclass correlation coefficients (ICCs); 95% confidence intervals were also calculated using a two-way random effects model. Continuous variables were assessed for normality using the Shapiro–Wilk test. All data were normally distributed; thus, parametric tests were performed. Continuous variables are presented as means ± standard deviations. Analysis of variance and Fisher’s exact test were used to compare independent groups. All comparative analyses were two-tailed, and *p*-values < 0.05 were considered statistically significant. All analyses were conducted using SPSS software (version 26.0; IBM Corp., Armonk, NY, USA). All analyses were conducted by statistician Eun Ae Jung (Soonchunhyang University, Gyeonggi-do, Republic of Korea)

## 3. Results

With respect to intraobserver reliability, all ICC values for Observers 1 and 2 were at least moderate according to the classification established by Koo et al. [36]. However, for Observer 3, the ICC values of the distal metatarsal articular angle and proximal phalangeal articular angle were < 0.5 on both sides. For all observers, the right and left distal metatarsal articular angles had the lowest ICC values. All sesamoid subluxation kappa values were above the threshold for good observer agreement (0.60) suggested by Landis et al. [37]. In terms of joint congruency, Observers 2 and 3 had kappa values exceeding the threshold for good agreement, whereas Observer 1 did not (Table 3). 

In terms of interobserver reliability, most measurements showed ICC values that were at least moderate. However, the ICCs of the intermetatarsal and distal metatarsal articular angles were <0.5 for both the first and second measurements. ICCs were similar for the first proximal phalangeal articular angle measurement. Among all measurements, the right and left distal metatarsal articular angles had the lowest ICC values for both the first and second measurements. With respect to the sesamoid subluxation grade, all kappa values were >0.8, indicating substantial concordance. Finally, all kappa values for joint congruency were <0.5 (Table 4).

We hypothesized whether there is a correlation between sesamoid subluxation grade and the value of each other angle. We also attempted to find out whether there is a correlation between sesamoid subluxation and whether or not there is joint congruency. As a result, sesamoid subluxation grade was shown to have a correlation with hallux valgus angle, intermetatarsal angle and joint congruency ( Table 5 and Table 6).

## 4. Discussion

In this study, the hallux valgus angle had the highest ICC, and most of the radiographic measurements had high ICC values on interobserver and intraobserver reliability testing. It means that most of the hallux valgus angle measurements show concordance and seem to have reliability. However, some of the items show low ICC values and less concordance. Specifically, the distal metatarsal angle had lower ICC values in the intraobserver reliability test and interobserver reliability test and much lower ICC values compared with previous study results [35,38]. Additionally, sesamoid subluxation grade has a correlation with the hallux valgus angle, the intermetatarsal angle and joint congruency.

Various radiographic measurements have been developed for the evaluation of hallux valgus. Multiple radiographic angles are used to assess the extent of deformity, determine whether the patient requires surgical or conservative treatment, select the type of surgery surgical intervention, and assess postoperative outcomes. In terms of surgical treatment, more than 100 procedures have been used for hallux valgus correction; no single operation can treat all hallux valgus deformities [6]. In addition, more recently, a minimally invasive procedure using an intramedullary nail device (MIIND) has been used for hallux valgus deformity correction, and Carlo Biz et al. reported that the intermetatarsal angle and hallux valgus angle significantly decreased after operative intervention using MIIND [39]. The surgical procedure should be carefully selected based on symptoms and preoperative radiological measurements. The use of those radiologic angle measurements in various areas is based on the reliance that they have reproducibility and reliability and provide a constant value for comparison. It is a major issue if those measurements are accurate, reliable, and repeatable or not [40]. The most widely performed measurements are the hallux valgus angle, the intermetatarsal angle, the proximal phalangeal metatarsal angle and the distal metatarsal articular angle. To date, using standardized weight-bearing plain radiographic images is considered the gold standard for the assessment of hallux valgus foot [41]. The traditional measuring methods of angles in plain radiographs include using a marking pen, pointing to the reference area, identifying each bone’s axis, and, finally, measuring angles with the goniometer. However, this approach is associated with intraobserver and interobserver errors, particularly with respect to the distal metatarsal articular angle. Moreover, it can be difficult to identify the articular surface [22,42]. Robinson et al. reported that the distal metatarsal articular angle considerably varied with the axial rotation of the first metatarsal bone, suggesting that measurements of the distal metatarsal angle on foot plain radiographs are susceptible to error [42]. It seems that such technical and structural limitations of plain radiographs are also related to the fact that distal metatarsal articular angle’s ICC has exceptionally low value in our study. So, in order to solve such limitations, the opinion to use foot weight-bearing CT for the diagnosis of hallux valgus was first reported in 2013 [32]. Many subsequent studies have described the use of weight-bearing CT to diagnose hallux valgus and facilitate treatment decision-making. However, in reality, it is difficult for clinicians to use foot-weight-bearing CT because of various limitations. First of all, hospitals equipped with equipment and facilities for foot weight-bearing CT are rare, the cost is relatively expensive when implemented, and there are issues regarding radiation exposure. Above all, it is not easy to know before the examination whether a hallux valgus patients are severe enough to operate. Therefore, weight-bearing CT is not recommended for routine examination of the hallux valgus foot. In addition, a foot plain radiograph is not only worth screening in terms of cost-effectiveness but also can be the only alternative in determining a treatment direction and a surgical treatment method in an environment where foot weight-bearing CT is not present. The errors that may occur with conventional radiographic measurements could be avoided by standardizing the technique used for the acquisition of weight-bearing plain radiographs and using specific reference points [43]. For example, the American Orthopaedic Foot and Ankle Society reference points for the metaphyseal–diaphyseal junction of the first and second metatarsal bones, as well as the proximal phalanx bone, are commonly used.

Talbot et al. reported that sesamoid subluxation beginning at the first metatarsal head is indicative of hallux valgus deformity [44]. The fact that during the progression of hallux valgus deformity, the head of the first metatarsal bone drifts away to the medial side from the sesamoids is widely understood, whereas the sesamoid bone maintains its anatomical relationship with the second metatarsal bone [45,46]. The adductor hallucis tendon, which has an insertion site in the base of the proximal phalanx and lateral sesamoid bones, stabilizes the sesamoid complex. The distance between the lateral sesamoid bone and the second metatarsal bone tends to remain constant, regardless of hallux valgus deformity [44,47]. Attempts have been made to analyze the relationship between the degree of medial subluxation of the sesamoid bone and the severity of hallux valgus deformity [9,48]. Traditionally, a foot tangential or axial sesamoid view radiograph has been suggested to be obtained to assess the amount of sesamoid subluxation. Particularly in cases of congruent hallux valgus feet, the sesamoid bone can seem subluxated on the weight-bearing plain radiograph yet be anatomically reduced in its facets. Kuwano et al. compared the rotation angle of the sesamoid bone between tangential sesamoid and plain radiographs; they found that plain radiographs were not appropriate for efforts to determine the grade of sesamoid subluxation [49]. Yildrim et al. demonstrated that the severity of sesamoid subluxation was inversely related to the degree of metatarsophalangeal joint dorsiflexion; they also found that measurements made on tangential sesamoid images were unreliable [50]. The recently reported study showed little difference between the two different radiograph views and also showed a significant correlation in the sesamoid bone position [26]. Based on these results, standard weight-bearing plain radiographs were used in the present study.

To date, there have been several articles that have studied how to measure each angle or what those angles mean. However, some may not be necessary, and their relationships have not been well established. We, therefore, investigated the reliabilities of radiographic measurements and correlations of sesamoid subluxation with the hallux valgus angle, the intermetatarsal angle and joint congruency to determine which radiographic measurements predicted the severity and prognosis of hallux valgus. The correlation of the sesamoid subluxation with the intermetatarsal angle was shown in this study. Lee et al*. reported that the intermetatarsal angle correlated with the sesamoid rotation angle [51]. From this correlation between radiographic measurements, we assumed valgus and pronation occur concurrently at the first tarsometatarsal joint. If this is true, the proximal metatarsal osteotomy should include a rotational component and reduce the intermetatarsal angle [22]. Okutda et al. argued postoperative sesamoid positions were important in the surgical outcome [28]. It is important to determine the sesamoid subluxation in preoperative evaluation. According to our study, the sesamoid subluxation grade will be helpful for the preoperative evaluation of a hallux valgus deformity.

This study has several limitations. Although several observers conducted this study, there was only one radiographic program, leading to some bias in measurements, and the results may lack generalizability. Although the number of patients involved in this study is quite large, the patient group is limited to only one institution. In the future, it is considered that a study to prove the results of this study is needed, using various imaging programs targeting patients in various institutions.

## 5. Conclusions

Measurements of the hallux valgus angle, interphalangeal angle, and joint congruency exhibited high interobserver and intraobserver reliabilities in this study; they also showed correlations with sesamoid subluxation grade. These measurements are important in cases of hallux valgus deformity, particularly because the sesamoid subluxation grade reflects the severity and prognosis of hallux valgus.

## Figures and Tables

**Figure 1 medicina-59-00876-f001:**
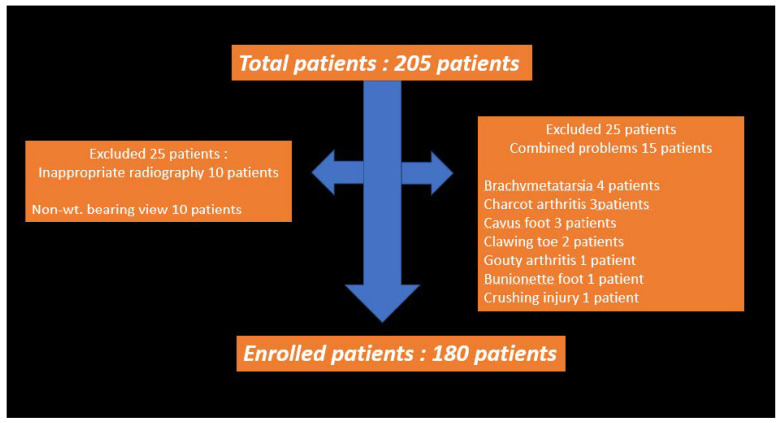
Summary of Enrolled patients: Total patients (n = 205)/10 patients excluded because of lack of weight-bearing scan/15 patients excluded because of: Brachymetatarsia (n = 4)/Charcot arthritis (n = 3)/Cavus foot deformity (n = 3)/Claw toe deformity (n = 2)/Gouty arthritis (n = 1)/Bunionette deformity (n = 1)/Crushing injury (n = 1)/Enrolled patients (n = 180).

**Figure 2 medicina-59-00876-f002:**
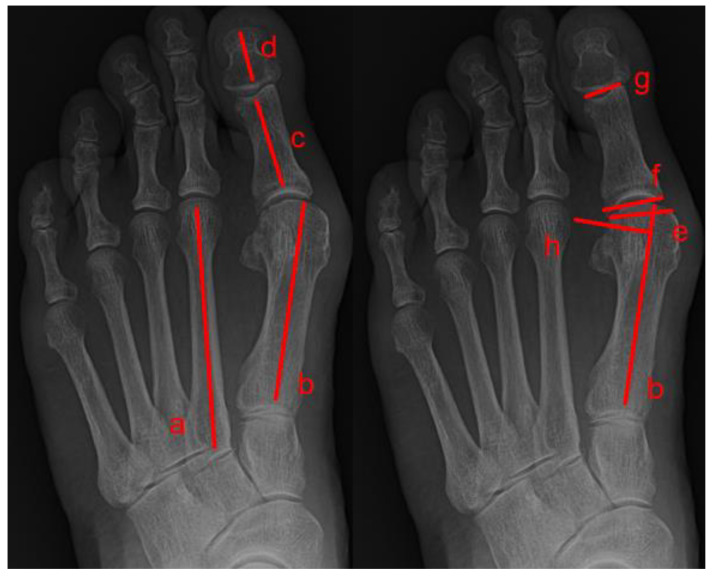
Angles between lines were measured on weight-bearing dorsoplantar foot radiographs. Line a is the longitudinal axis of the second metatarsal, line b is the longitudinal axis of the first metatarsal, line c is the longitudinal axis of the proximal phalanx, line d is the longitudinal axis of the distal phalanx, line e is the orientation of the first metatarsal distal articular surface, line f is the orientation of the proximal phalangeal base articular surface, line g is the orientation of the proximal phalangeal distal articular surface, and line h is the perpendicular axis of the longitudinal axis of the first metatarsal. The hallux valgus angle is the angle between b and c, and the intermetatarsal angle is the angle between a and b, the hallux interphalangeal angle is the angle between c and d, the proximal phalangeal articular angle is the angle between f and g, and the distal metatarsal articular angle is the angle between h and f.

**Figure 3 medicina-59-00876-f003:**
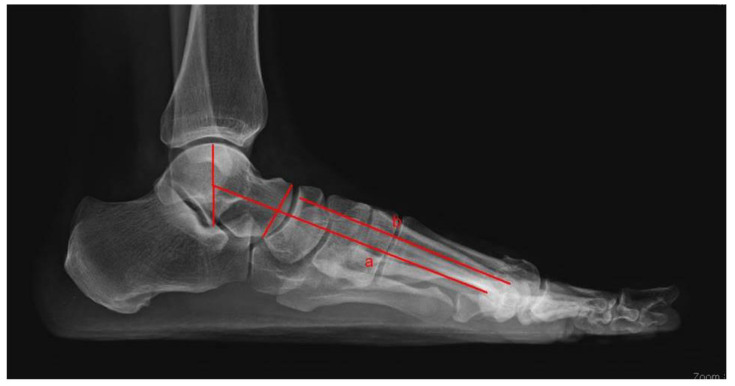
Angles between lines were measured on weight-bearing lateral foot radiographs. Line a is the lateral central axis of the talus, and line b is the longitudinal axis of the first metatarsal. The tarso-first metatarsal angle (Meary’s angle) is the angle between a and b.

**Table 1 medicina-59-00876-t001:** Patient clinical and demographic characteristics.

Total	205
Excluded patients	25
Patient number	180
Age	52.78
Male:Female	18:162
Operated feet	240
Operated sided Right:Left:Bilateral	54:66:60

**Table 2 medicina-59-00876-t002:** Sesamoid subluxation grade.

Grade	
0	No sesamoid lateral displacement
1	Sesamoid overlap < 25%
2	25% < Sesamoid overlap < 50%
3	50% < Sesamoid overlap < 75%
4	75% < Sesamoid overlap

**Table 3 medicina-59-00876-t003:** Intraobserver reliability data of Observers 1–3.

	Observer1			Observer2			Observer3		
	CCC	95% CI		CCC	95% CI		CCC	95% CI	
R HVA	0.890	(0.844	0.923)	0.989	(0.984	0.992)	0.649	(0.530	0.743)
R IMA	0.810	(0.736	0.865)	0.909	(0.872	0.936)	0.756	(0.669	0.823)
R DMAA	0.650	(0.535	0.741)	0.842	(0.779	0.888)	0.306	(0.135	0.460)
R T-MA	0.875	(0.824	0.913)	0.974	(0.962	0.982)	0.541	(0.411	0.650)
R Ppa	0.734	(0.637	0.808)	0.922	(0.889	0.945)	0.396	(0.238	0.533)
R HIA	0.804	(0.731	0.859)	0.924	(0.892	0.947)	0.734	(0.638	0.807)
L HVA	0.964	(0.950	0.975)	0.986	(0.981	0.990)	0.945	(0.922	0.961)
L IMA	0.154	(0.052	0.253)	0.950	(0.930	0.964)	0.704	(0.605	0.782)
L DMAA	0.611	(0.494	0.706)	0.860	(0.806	0.899)	0.335	(0.177	0.476)
L T-Ma	0.864	(0.813	0.902)	0.979	(0.970	0.985)	0.554	(0.435	0.653)
L Ppa	0.827	(0.764	0.874)	0.908	(0.873	0.935)	0.395	(0.266	0.510)
L HIA	0.754	(0.671	0.819)	0.932	(0.904	0.951)	0.503	(0.361	0.623)
CCC: Concordance correlation coefficient									
	Observer1			Observer2			Observer3		
	Weighted Kappa	95% CI		Weighted Kappa	95% CI		Weighted Kappa	95% CI	
R Sesa	0.647	(0.515	0.778)	0.924	(0.874	0.974)	1.000	(1.000	1.000)
L Sesa	0.692	(0.583	0.800)	0.966	(0.932	1.000)	0.994	(0.982	1.000)
	Observer1			Observer2			Observer3		
	Kappa	95% CI		Kappa	95% CI		Kappa	95% CI	
R Cong	0.581	(0.378	0.784)	0.936	(0.864	1.000)	0.939	(0.856	1.000)
L Cong	0.468	(0.285	0.651)	0.937	(0.866	1.000)	1.000	(1.000	1.000)

CI, confidence interval; CCC, concordance correlation coefficient; R, right side; L, left side; HVA, hallux valgus angle; IMA, intermetatarsal angle; DMAA, distal metatarsal articular angle; T-MA, tarso-first metatarsal angle; PPAA, proximal phalangeal articular angle; HIA, hallux interphalangeal; SSG, sesamoid subluxation grade.

**Table 4 medicina-59-00876-t004:** Interobserver reliability of Observers 1–3 for the first and second measurements.

	First			Second		
	CCC	95% CI		CCC	95% CI	
R HVA	0.747	(0.591	0.843)	0.867	(0.810	0.907)
R IMA	0.377	(0.271	0.481)	0.407	(0.318	0.507)
R DMAA	0.285	(0.183	0.411)	0.189	(0.099	0.320)
R T-MA	0.640	(0.552	0.718)	0.611	(0.528	0.690)
R Ppa	0.531	(0.393	0.634)	0.644	(0.521	0.727)
R HIA	0.656	(0.551	0.774)	0.773	(0.692	0.843)
L HVA	0.861	(0.785	0.898)	0.866	(0.802	0.903)
L IMA	0.131	(0.020	0.542)	0.539	(0.424	0.623)
L DMAA	0.322	(0.237	0.424)	0.206	(0.101	0.320)
L T-Ma	0.667	(0.588	0.740)	0.575	(0.485	0.663)
L Ppa	0.489	(0.369	0.660)	0.752	(0.664	0.853)
L HIA	0.591	(0.500	0.674)	0.675	(0.599	0.753)
CCC: Concordance correlation coefficient						
	first		second			
	Kendall’s Coefficient	*p*-value	Kendall’s Coefficient	*p*-value		
R Sesa	0.890	<0.0001	0.846	<0.0001		
L Sesa	0.817	<0.0001	0.819	<0.0001		
	1st			2nd		
	Kappa	95% CI		Kappa	95% CI	
R Cong	0.412	(0.306	0.519)	0.393	(0.286	0.499)
L Cong	0.501	(0.400	0.602)	0.483	(0.382	0.584)

CI, confidence interval; CCC, concordance correlation coefficient; R, right side; L, left side; HVA, hallux valgus angle; IMA, intermetatarsal angle; DMAA, distal metatarsal articular angle; T-MA, tarso-first metatarsal angle; PPAA, proximal phalangeal articular angle; HIA, hallux interphalangeal; SSG, sesamoid subluxation grade.

**Table 5 medicina-59-00876-t005:** Correlations of sesamoid subluxation with radiological angles and joint congruency for Observer 1.

	Sesamoid Subluxation										*p*-Value ^†^
	0		1		2		3		4		
Patients, n	1		5		18		19		70		
HVA	16.40		20.40	±4.62	24.49	±6.37	23.36	±5.45	33.39	±9.40	<0.0001
IMA	4.40		10.22	±1.97	11.14	±2.24	11.24	±1.81	14.52	±2.61	<0.0001
DMAA	8.00		15.36	±9.78	11.90	±6.13	10.91	±5.13	12.81	±5.80	0.4906
T-MA	5.90		5.92	±10.69	4.74	±4.81	0.90	±6.51	0.72	±6.96	0.1122
Ppa	14.20		5.02	±3.53	7.56	±4.01	8.24	±4.29	6.80	±4.27	0.2065
HIA	19.20		11.24	±4.90	10.83	±4.18	9.78	±3.46	7.19	±4.74	0.0009
Cong											<0.0001
C	1	(100.0%)	4	(80.0%)	4	(22.2%)	7	(36.8%)	5	(7.1%)	
N	0	(0.0%)	1	(20.0%)	14	(77.8%)	12	(63.2%)	65	(92.9%)	

^†^ Analysis of variance or Fisher’s exact test.

**Table 6 medicina-59-00876-t006:** Correlations of sesamoid subluxation with radiological angles and joint congruency for Observer 2.

	Sesamoid Subluxation										*p*-Value ^†^
	0		1		2		3		4		
Patients, n	1		5		18		20		70		
HVA	15.30		15.44	±7.15	22.91	±6.86	23.07	±4.18	32.59	±8.94	<0.0001
IMA	5.60		9.14	±3.43	10.60	±1.92	11.04	±2.12	14.82	±2.53	<0.0001
DMAA	0.90		10.02	±7.11	8.83	±7.05	9.85	±4.91	11.42	±5.83	0.2089
T-MA	16.80		5.08	±6.17	3.64	±7.11	1.41	±6.25	0.56	±7.29	0.0710
Ppa	6.40		9.80	±3.40	7.16	±4.27	8.74	±3.97	6.85	±4.35	0.3061
HIA	15.10		13.82	±4.99	11.24	±4.92	11.12	±3.01	8.48	±4.96	0.0107
Cong											<0.0001
C	1	(100.0%)	5	(100.0%)	10	(55.6%)	13	(65.0%)	5	(7.1%)	
N	0	(0.0%)	0	(0.0%)	8	(44.4%)	7	(35.0%)	65	(92.9%)	

^†^ Analysis of variance or Fisher’s exact test. HVA, hallux valgus angle; IMA, intermetatarsal angle; DMAA, distal metatarsal articular angle; T-MA, tarso-first metatarsal angle; PPAA, proximal phalangeal articular angle; HIA, hallux interphalangeal.

## Data Availability

Data sharing is not applicable to this article because no datasets were made or analyzed during this study.

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
