# Peer review of "Correlations of Sesamoid Bone Subluxation with the Radiologic Measures of Hallux Valgus and Its Clinical Implications"

_medicina, 2023, doi:10.3390/medicina59050876_

Round 1

Reviewer 1 Report

I suggest the authors to remove the inter-observer reliability analysis, to make this study focus on the sesamoid bone subluxation. One reliable single-examiner is sufficient to draw a conclusion.

Author Response

First of all, thank you for your kind and thoughtful review. We also thought about this matter before submitting the article. However, it was decided that it was necessary to check the inter-observer reliability in order to differentiate it from other papers on similar topics. If you are not uncomfortable with it, we would like to keep it without deleting it from the paper.

King regards,

Reviewer 2 Report

I congratulate the authors on an ambitious retrospectively study. The research is robust and the design well considered. I look forward to seeing the end result of this work when it is finally complete and published. I commend the authors for their work - both all of the work leading up to this point and for the planning of this investigation - their contribution to the hallux valgus and sesamoid bone subluxation literature. I do have some comments about certain methodological issues covered below under MAJOR ISSUES the majority of which are related to clarity of the work as it is currently written.

TITLE

The title should be amended slightly to ensure that the reader understands the type of research immediately that this paper for clarity, interes and ease of read.

ABSTRACT

It is hard to get the detail in an abstract when the word count is limited and this is often the hardest part of a paper to write. However, I do feel that it would be beneficial to explain what specifically you are looking at in relation to assesment and halliux valgus and sesamoid bone subluxation(this also applies to the main body of the paper). Is it the development of hallux valgus associated with poor diagnostic . This needs to be made clearer throughout the paper

KEYWORDS:

Please use recognised MeSH terms as this will assist others when they are searching for information on your research topic. The following website will provide these (simply start typing in a keyword and see if it exists or find an alternative if it does not): https://www.ncbi.nlm.nih.gov/mesh

The introduction is weak. An introduction should announce your topic, provide context and a rationale for your work, while catching the reader´s interest and attention. The above has not been given in the introduction that I have read. Thus, I suggest in this section should be improved, with more details about prevalence, impact related with flat feet associated with the impact of the quality of health. It is indeed important paper but it lacks several critical references, in which it was presented related with this condition, and it should be emphasized in the INTRODUCTION or Discussion of the authors' paper. More info info in the research of Perez Boa  et al entitled Geometry of the Proximal Phalanx of Hallux and First Metatarsal Bone to Predict Hallux Abducto Valgus: A Radiological Study https://pubmed.ncbi.nlm.nih.gov/27861517/. Furthermore, in relation with radiographic analysis on the distortion of the anatomy of first metatarsal head in dorsoplantar projection https://pubmed.ncbi.nlm.nih.gov/32748863/

Also, please describe the hypothesis in this section.

MATERIAL AND METHODS:

This section are appropriate and described in adequate detail while the conclusions clearly link to the data presented. Please, expand and clarification information related with this research for adhere to reporting STROBE guidelines.

RESULTS:

The results section is very appropriate according to the developed methods and the journal´s scope. In the table add the p values. Also, to update the figure and all tables should be inserted into the main text close to their first citation and must be numbered following their number of appearance with  should have a short explanatory title and caption.

Furthemore, all tables columns should have an explanatory heading. To facilitate the copy-editing of larger tables, smaller fonts may be used, but no less than 8 pt. in size. Authors should use the Table option of Microsoft Word to create tables. More info in Preparing Figures, Schemes and Tables https://www.mdpi.com/journal/medicina/instructions

DISCUSSION:

Include this section the principal strengths and weaknesses in relation to other studies, discussing important differences in results; the meaning of the study: possible explanations and implications and unanswered questions and future research

CONCLUSION:

summarize the conclusions in order to reflect only the study findings.

Author Response

First of all, thank you for your kind and thoughtful review. Thanks to your comments, our article was able to develop in a better direction. We will explain the contents item by item.

TITLE

The title should be amended slightly to ensure that the reader understands the type of research immediately that this paper for clarity, interes and ease of read.

  • Thank you for your nice comments. After listening to your opinion, I thought about it for a long time, but I couldn't come up with a more attractive title that would contain all the content of this paper. If you have any good ideas, we will actively reflect them.

ABSTRACT

It is hard to get the detail in an abstract when the word count is limited and this is often the hardest part of a paper to write. However, I do feel that it would be beneficial to explain what specifically you are looking at in relation to assesment and halliux valgus and sesamoid bone subluxation(this also applies to the main body of the paper). Is it the development of hallux valgus associated with poor diagnostic. This needs to be made clearer throughout the paper

  • We tried to figure out the relationship between hallux valgus angle, interphalangeal angle, and joint congruency and sesamoid subluxation grade. The reason is that sesamoid subluxation grade is related to the severity and prognosis of hallux valgus. Finally, we revealed the relationship between hallux valgus angle, interphalangeal angle, and joint congruency and severity and prognosis of hallux valgus in this paper. The contents were described in the conclusion of the main body, and added to the abstract according to your opinion.

KEYWORDS:

 Please use recognised MeSH terms as this will assist others when they are searching for information on your research topic. The following website will provide these (simply start typing in a keyword and see if it exists or find an alternative if it does not): https://www.ncbi.nlm.nih.gov/mesh

  • We tried to replace the words in the keywords with other words according to your comment, but couldn't find the right words. I'd really appreciate it if you could tell me any good ideas you know. 

The introduction is weak. An introduction should announce your topic, provide context and a rationale for your work, while catching the reader´s interest and attention. The above has not been given in the introduction that I have read.

 Thus, I suggest in this section should be improved, with more details about prevalence, impact related with flat feet associated with the impact of the quality of health. It is indeed important paper but it lacks several critical references, in which it was presented related with this condition, and it should be emphasized in the INTRODUCTION or Discussion of the authors' paper. More info info in the research of Perez Boa  et al entitled Geometry of the Proximal Phalanx of Hallux and First Metatarsal Bone to Predict Hallux Abducto Valgus: A Radiological Study https://pubmed.ncbi.nlm.nih.gov/27861517/. Furthermore, in relation with radiographic analysis on the distortion of the anatomy of first metatarsal head in dorsoplantar projection https://pubmed.ncbi.nlm.nih.gov/32748863/

  • First of all, thank you very much for recommending two really good papers. We read carefully the papers and cited both in our paper. Also, as you said, I've added things about prevalence and flat foot.

Also, please describe the hypothesis in this section.

  • We added that content in introduction

(We assumed that they might have a relationship with each other.)

MATERIAL AND METHODS:

This section are appropriate and described in adequate detail while the conclusions clearly link to the data presented. Please, expand and clarification information related with this research for adhere to reporting STROBE guidelines.

 https://www.ncbi.nlm.nih.gov/pmc/articles/PMC6398292/

  • As you said, we read STROBE guidelines. If you tell me if we can send it to you or if you want us to add it to the text, We will.

RESULTS:

The results section is very appropriate according to the developed methods and the journal´s scope. In the table add the p values. Also, to update the figure and all tables should be inserted into the main text close to their first citation and must be numbered following their number of appearance with  should have a short explanatory title and caption.

  • Thanks for your kind comments. First of all, the p value was described in the text, but we would like to ask if it should be written in the table again. And both figures and tables are located as close as possible to the main text with short explanations. If you have any better ideas, please let me know.

Furthemore, all tables columns should have an explanatory heading. To facilitate the copy-editing of larger tables, smaller fonts may be used, but no less than 8 pt. in size. Authors should use the Table option of Microsoft Word to create tables. More info in Preparing Figures, Schemes and Tables https://www.mdpi.com/journal/medicina/instructions

  • The font size is an appropriately small font of 8 or more, and we make increased the table size.

DISCUSSION:

Include this section the principal strengths and weaknesses in relation to other studies, discussing important differences in results; the meaning of the study: possible explanations and implications and unanswered questions and future research

  • Compared to other papers, we emphasized the reliability of using multiple observers as an advantage. In addition, limitations and future research contents have already been described in the limitation part. If you have any additional points to add, we will actively reflect them.

CONCLUSION:

summarize the conclusions in order to reflect only the study findings.

  • Based on your opinion, we have removed the content that we consider useless.

Reviewer 3 Report

Many thanks to the authors for having presented a so interesting study about “Correlations of Sesamoid Bone Subluxation with the Hallux Valgus Angle, Intermetatarsal Angle, and Metatarsophalangeal Joint Congruency in Hallux Valgus Patients”.

Please before resubmitting the revision version of the article, read the editorial rules carefully, and check for other editorial aspects, such as: text alignment, text justification at the head, etc.

The language is very good, the manuscript doesn’t need correction by a person of English mother tongue.

ABSTRACT

The abstract is complete and contains the main results of the study and the most important considerations.

Please divide it into sections, reflecting the structure of the article.

KEY WORDS

Please provide them in alphabetical order.

BACKGROUND

The introduction is well structured.

Please add some possible causes if hallux valgus recurrence after corrective surgery, as the topic is only briefly cited.

The biomechanics and pathophysiology are very clearly explained.

Line 45: Abnormal alignment and structure of the first metatarsophalangeal joint would contribute to the collapse of the medial longitudinal arch of the foot in hallux valgus patients [6]. However, please add some considerations about the role of hypermobility of the first ray in HV aetiology, both in normal people and in athletes such as ballet dancer, quoting relative references.

Also, an overview of the options available for surgical and conservative treatment should be added. For example, cite:

·         Wülker N, Mittag F. The treatment of hallux valgus. Dtsch Arztebl Int. 2012 Dec;109(49):857-67; quiz 868. doi: 10.3238/arztebl.2012.0857. Epub 2012 Dec 7. PMID: 23267411; PMCID: PMC3528062

The purpose of the article is clearly stated.

Line 60: Percutaneous approaches and minimally invasive surgery are increasingly used because these approaches can achieve results that are at least equivalent to the results of conventional open surgery and have lower complications [11]. Please, quote also:

·         Functional and radiographic outcomes of hallux valgus correction by mini-invasive surgery with Reverdin-Isham and Akin percutaneous osteotomies: a longitudinal prospective study with a 48-month follow-up. J Orthop Surg Res. 2016 Dec 5;11(1):157. doi: 10.1186/s13018-016-0491-x.

MATERIAL AND METHODS

This section contains enough information to understand and possibly repeat the study.

Please rewrite the classification of sesamoid subluxation in a more schematic way; it would be useful to add a table on this topic.

STATISTICAL ANALYSIS

The methods used for the statistical analysis are well detailed.

Also, add who performed the statistical analysis, if an independent statistician or the same authors.

RESULTS

The results presented are complete and reflect the MM section.

DISCUSSION

The content of this section is coherent with the rest of the paper and adequately backed up by previous literature.

The limitations of the study are explained clearly.

Line 278: In terms of surgical treatment, more than 100 procedures have been used for hallux valgus correction; no single operation can treat all hallux valgus deformities [2]. Please discuss also the result reported by MIIND for HV correction in relation to different angle values and their improving after surgery.

Line 291: “this approach is associated with intraobserver and interobserver errors”. Please expand this concept, providing possible explanations about the sources of these errors.

CONCLUSIONS

The conclusions reflect and refer to the results of the study.

REFERENCES

The references are up to date; however, please remove the citations predating 2010 if not necessary.

Also, please integrate them as previously suggested.

TABLES AND FIGURES

The tables and figures are clear and pertinent to the content of the paper; please integrate them as suggested in the previous sections.

Please add more radiographical images regarding the measurements: ideally, there should be a different image for every measurement, to facilitate understanding.

Author Response

First of all, thank you for your kind and thoughtful review. Thanks to your comments, our thesis was able to develop in a better direction. We will explain the contents item by item.

ABSTRACT

The abstract is complete and contains the main results of the study and the most important considerations.

Please divide it into sections, reflecting the structure of the article.

=>Thanks for your advices. We divided abstract into 4 section, adding contents about material and method, conclusion.

KEY WORDS

Please provide them in alphabetical order.

=>we changed keywords in alphabetical order

BACKGROUND

The introduction is well structured.

Please add some possible causes if hallux valgus recurrence after corrective surgery, as the topic is only briefly cited.

=>According to your advice, we add possible cause about hallux valgus recurrence citing related article

The causes of recurrence are not specific and thought to be multifactorial, including both surgeon-triggered factors and patients related factors.

-> The causes of recurrence are thought to be multifactorial, including surgical factors such as choice of the appropriate procedure and technical competency and patients related factors such as anatomic predisposition, medical comorbidities, compliance with post correction instructions.

The biomechanics and pathophysiology are very clearly explained.

Line 45: Abnormal alignment and structure of the first metatarsophalangeal joint would contribute to the collapse of the medial longitudinal arch of the foot in hallux valgus patients [6]. However, please add some considerations about the role of hypermobility of the first ray in HV aetiology, both in normal people and in athletes such as ballet dancer, quoting relative references.

  • ??

Also, an overview of the options available for surgical and conservative treatment should be added. For example, cite:

  • Wülker N, Mittag F. The treatment of hallux valgus. Dtsch Arztebl Int. 2012 Dec;109(49):857-67; quiz 868. doi: 10.3238/arztebl.2012.0857. Epub 2012 Dec 7. PMID: 23267411; PMCID: PMC3528062

-> we added following sentences citing your recommended article

Treatment options for hallux valgus deformity comprise surgical treatment and conservative treatment

The purpose of the article is clearly stated.

Line 60: Percutaneous approaches and minimally invasive surgery are increasingly used because these approaches can achieve results that are at least equivalent to the results of conventional open surgery and have lower complications [11]. Please, quote also:

  • Functional and radiographic outcomes of hallux valgus correction by mini-invasive surgery with Reverdin-Isham and Akin percutaneous osteotomies: a longitudinal prospective study with a 48-month follow-up. J Orthop Surg Res. 2016 Dec 5;11(1):157. doi: 10.1186/s13018-016-0491-x.
  • We quoted your recommended article and added following sentences.
  • Percutaneous approaches and minimally invasive surgery are increasingly used because these approaches can achieve results that are at least equivalent to the results of conventional open surgery and have lower complications

MATERIAL AND METHODS

This section contains enough information to understand and possibly repeat the study.

Please rewrite the classification of sesamoid subluxation in a more schematic way; it would be useful to add a table on this topic.

  • Thanks for your advice, we added table rewriting the classification of sesamoid subluxation in a more schematic way

STATISTICAL ANALYSIS

The methods used for the statistical analysis are well detailed.

Also, add who performed the statistical analysis, if an independent statistician or the same authors.

  • We added who performed the statistical analysis following your advice

RESULTS

The results presented are complete and reflect the MM section.

DISCUSSION

The content of this section is coherent with the rest of the paper and adequately backed up by previous literature.

The limitations of the study are explained clearly.

Line 278: In terms of surgical treatment, more than 100 procedures have been used for hallux valgus correction; no single operation can treat all hallux valgus deformities [2]. Please discuss also the result reported by MIIND for HV correction in relation to different angle values and their improving after surgery.

  • We added result about MIIND citing related study

Also, recently minimal invasive procedure using intramedullary nail device(MIIND) is used for hallux valgus deformity correction and Carlo biz et al reported intermetatarsal angle, hallux valgus angle significantly decreased after operative intervention using MIIND

Line 291: “this approach is associated with intraobserver and interobserver errors”. Please expand this concept, providing possible explanations about the sources of these errors.

  • We consider the source of such errors to be human error. It's hard for us to think of anything that can be further explained on that part, so if you have any good ideas please share to us. We are ready to actively reflect such opinions.

CONCLUSIONS

The conclusions reflect and refer to the results of the study.

REFERENCES

The references are up to date; however, please remove the citations predating 2010 if not necessary.

Also, please integrate them as previously suggested.

  • We considered removing citations predating 2010, but as it would go track off from our original thesis, so in order to maintain a logical flow we decided to maintain the current citation  

TABLES AND FIGURES

The tables and figures are clear and pertinent to the content of the paper; please integrate them as suggested in the previous sections.

Please add more radiographical images regarding the measurements: ideally, there should be a different image for every measurement, to facilitate understanding.

  • To facilitate understanding, we divided figure2 into 2 measuring image(axis about phalanges, metatarsal bone and axis oblique to angle of phalanges, metatarsal bone)

Round 2

Reviewer 1 Report

The comments have been addressed well.

Author Response

We are very glad to hear your kind comments

Thank you

King regards,

Reviewer 2 Report

I would like to thank the authors for their work, however I did not feel the authors made any significant improvements with regards to the main issues I raised in the first review. In its present state the paper provides no clear evidence that the authors propose a experiment must have been conducted rigorously. In addition, the clinical relevance is not high and need to be given rationale.

The authors still need to make substantial changes inline with the issues raised in the first review. 

Indeed, the authors need respond all questions according to all the required recommendations and add the modifications to the manuscript text with yellow highlight.

Reviewer 3 Report

The authors answered my comments properly, impruving the manuscript quality: well done!

Author Response

(The authors gave the same response as above.)

Round 3

Reviewer 2 Report

The authors have clearly and adequately addressed all comments raised by the reviewers. Please also consider adding the p value in the table 1 and  remove the colour for improve the quality to the presentantion and  correct the error in the term "205 PaItents"  by "205 patients" in the figure 1